# Applications of Digital Twin across Industries: A Review



**Maulshree Singh** [1] , **Rupal Srivastava** [1,2,†] , **Evert Fuenmayor** [1,†] , **Vladimir Kuts** [2,3,4] , **Yuansong Qiao** [3] , **Niall Murray** [3] **and Declan Devine** [1,*]

1 PRISM Research Institute, Technological University of the Shannon, Midlands Midwest, Athlone Main Campus, N37 HD68 Athlone, Ireland; m.singh@research.ait.ie (M.S.); rsrivastava@ait.ie (R.S.); efuenmayor@ait.ie (E.F.)
2 Confirm Smart Manufacturing, Science Foundation Ireland, V94 C928 Limerick, Ireland; vladimir.kuts@ul.ie
3 Software Research Institute, Athlone Institute of Technology, Technological University of the Shannon, Midlands Midwest, N37 HD68 Athlone, Ireland; yuangsongqiao@ait.ie (Y.Q.); nmurray@ait.ie (N.M.)
4 Department of Electronics and Computer Engineering, University of Limerick, V94 T9PX Limerick, Ireland
* Correspondence: declan.devine@tus.ie; Tel.: +353-90-646-8291
† These authors contributed equally to this work.

**Abstract:** One of the most promising technologies that is driving digitalization in several industries is Digital Twin (DT). DT refers to the digital replica or model of any physical object (physical twin). What differentiates DT from simulation and other digital or CAD models is the automatic bidirectional exchange of data between digital and physical twins in real-time. The benefits of implementing DT in any sector include reduced operational costs and time, increased productivity, better decision making, improved predictive/preventive maintenance, etc. As a result, its implementation is expected to grow exponentially in the coming decades as, with the advent of Industry 4.0, products and systems have become more intelligent, relaying on collection and storing incremental amounts of data. Connecting that data effectively to DTs can open up many new opportunities and this paper explores different industrial sectors where the implementation of DT is taking advantage of these opportunities and how these opportunities are taking the industry forward. The paper covers the applications of DT in 13 different industries including the manufacturing, agriculture, education, construction, medicine, and retail, along with the industrial use case in these industries.

**Keywords:** Digital Twin; Industry 4.0; Smart Manufacturing; system optimization; predictive maintenance

## 1. Introduction

The term 'Digital Twin' (DT) has gained popularity recently in academic as well as industrial circles, even though it was conceptualized decades ago. In 2002, Michael Grieves, who introduced the concept of DT, defined it as *"A set of virtual information constructs that fully describes a potential or actual physical manufactured product from the micro atomic level to the macro geometrical level. At its optimum, any information that could be obtained from inspecting a physically manufactured product can be obtained from its Digital Twin"* [1]. When defined as such, a Digital Twin is comprised of three components (Figure 1):

(i) Physical twin: A real-world entity (living/non-living) such as part/product, machine, process, organization, or human, etc.
(ii) Digital twin: The digital representation of the physical twin with the capability to mimic/mirror its physical counterpart in real time.
(iii) Linking mechanism: The bidirectional flow of data between the two which operates automatically in real-time.

Though DT has been defined by academia and industry in several different ways, one thing that they all agree on is its benefits. It reduces operational costs and time, increases the productivity of the existing system, helps in the decision-making process, improves maintenance schedules and activities, provides remote access, makes a safer

working/operational environment, and promotes sustainability [2]. Owing to the varied advantages and applications of DTs, their adoption in different sectors has accelerated in recent years. As reported by Grand View Research, the global market of DT was estimated at USD 5.04 billion in 2020 and is expected to reach USD 86.09 billion by 2028, with a compound annual growth rate of 42.7% from 2021 to 2028 [3]. One of the reasons the demand for DTs has accelerated is the recent outbreak of the pandemic, COVID-19 [4,5]. Lockdowns due to the pandemic resulted in supply chain breakage, a shortage of workforce, and remote or non-contact working environment [6], due to which digitization and the advancement of processes with minimum human contact have gained much more importance [4]. According to the Gartner survey, 31% of the companies are using DT for the remote monitoring of assets to reduce the frequency of in-person monitoring, such as hospital patients and mining operations to increase the safety of employees and customers [7].

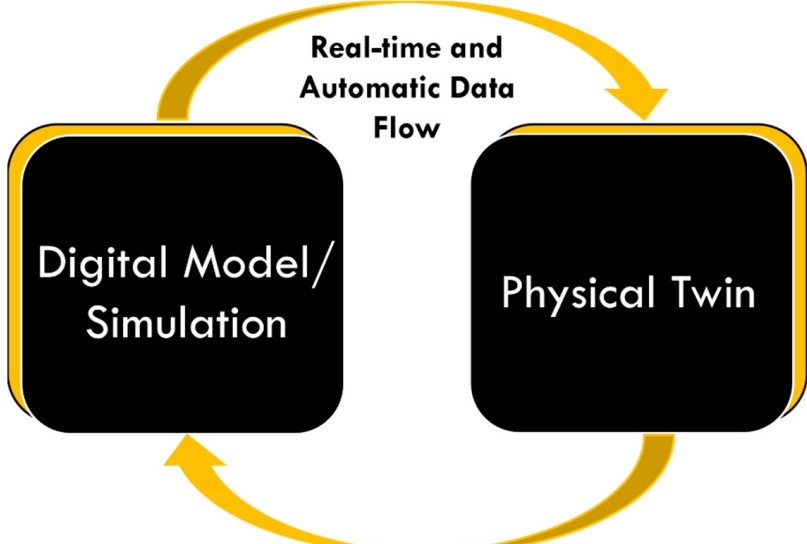

**Figure 1.** Block diagram of a Digital Twin.

DT technology has found its applications in different industries going through digital transformation. Martin and Nadja [8], in their study on DT applications, identified ten major industrial sectors where DT has been applied. These industries are (i) Aerospace, (ii) Manufacturing, (iii) Healthcare, (iv) Energy, (v) Automotive, (vi) Petroleum, (vii) Public sector, (viii) Mining, (ix) Marine, and (x) Agricultural. They also pointed out that DT in these industries has been used for three main purposes i.e., simulation, monitoring, and control. However, the applications of DT are not just limited to these three purposes only. DT now is used for design, validation, customization, optimization, prediction, and maintenance. According to Juniper Research, the industry which will be leading in DT deployment by 2021 is manufacturing (34%), followed by energy (18%) [9].

This paper provides a summary of the current applications of DT in different industries. To get a better understanding of DT, our previous review paper [2] can be referred to, which talks about the definitions of DT across the literature, its characteristics, types, advantages, and challenges. The DT applications that have been reported in the literature regularly are in the field of manufacturing, cyber-physical systems, or in the context of Industry 4.0 in general [10–23]. There are exceptions, but these review papers either cover only one industry [24–34] or only a few industries [4,35–38]. C. Semeraro et al., in their review of 115 papers on DT, found that the most explored DT application contexts are: healthcare, maritime and shipping, manufacturing, city management, and aerospace [39]. How the DT technology is being used in these five aforementioned sectors is discussed in this paper. The paper also extends the discussion of DT applications in the real world to sectors such as education, construction, and retail, areas not considered by Martin and Nadja [8] (Figure 2).

The aim of the paper is to identify and understand the potential of DT in any sector and the purpose of implementing it for the right application, which can prove to beneficial to any researcher, business, or sector before investing in the technology to unleash the true potential of the technology.

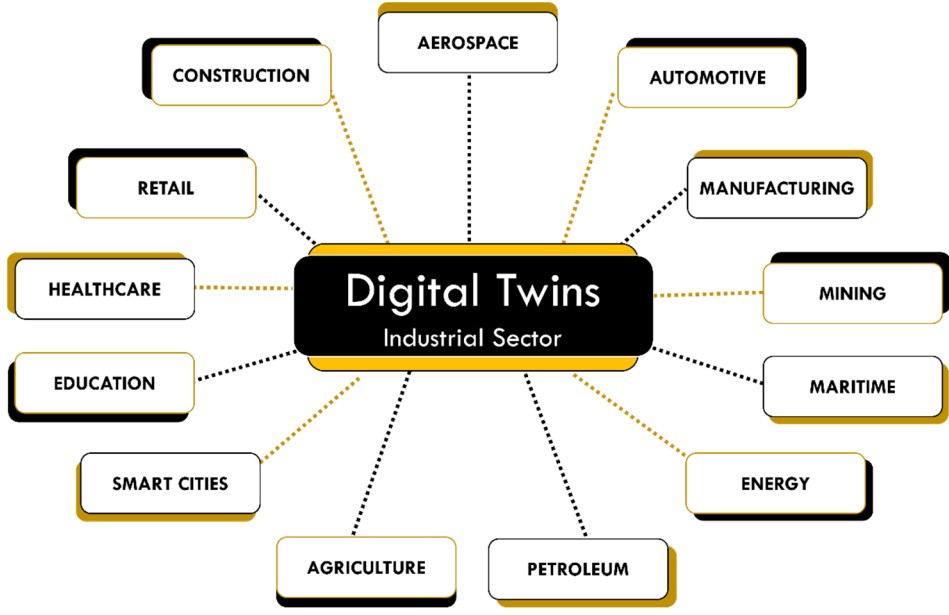

**Figure 2.** Different industrial sectors where DT has found its applications.

The number of studies in each sector discussed in the paper is varied, but all of them are increasing year by year. Table 1 shows the number of publications related to DT in a particular sector in Scopus, Science Direct, and Google Scholar (last checked May 2022). The increase in DT-related publications is quite recent, i.e., 2016 onwards, but since then the growth has been exponential [2]. The manufacturing industry is leading the chart with the most number of studies, followed by Energy, Construction, and Education, even though the applications of DT in the field of education has been relatively newer. The Aerospace sector, which is the pioneering industry in terms of DT technology, has not that many publications as compared to other sectors. The implementation of DT in sectors such as retail and agricultural is fairly new as compared to others, thus resulting in a low number of publications.

**Table 1.** Number of Digital Twin-related publications in each industry based on different platforms.

| Industry | No. of Publications | | |
|---|---|---|---|
| | Scopus | Science Direct | Google Scholar |
| **Aerospace and Aeronautics** | 168 | 980 | 2360 |
| **Manufacturing** | 1823 | 3295 | 25,300 |
| **Healthcare and Medicine** | 188 | 819 | 4810 |
| **Power Generation/Energy** | 780 | 2863 | 17,300 |
| **Automotive** | 177 | 939 | 7910 |
| **Oil and Gas** | 434 | 1463 | 9780 |
| **Smart City** | 236 | 462 | 4090 |
| **Mining** | 191 | 1001 | 9700 |
| **Maritime and Shipping** | 59 | 360 | 2330 |
| **Agricultural** | 63 | 495 | 3900 |
| **Education** | 223 | 939 | 14,900 |
| **Construction** | 748 | 1825 | 17,700 |
| **Retail** | 21 | 286 | 2830 |

## 2. Aerospace and Aeronautics

Aerospace/aeronautics fields were the pioneer areas where DT was explored first by NASA and by the U.S. Air Force [40–43]. The main applications of DTs in this industry include optimizing the performance and reliability of the space vehicle/aircraft, predicting and resolving maintenance issues, and making the missions safer for the crew. The main application of DT in this industry started with the intention to optimize the performance and reliability of the space vehicle/aircraft. According to the 2010 technological roadmap by NASA [44], the four applications of DT for them were:

(i)   Simulating the flight before the launch of the actual vehicle to maximize the mission success.
(ii)  Continuously mirroring the actual flight and updating the conditions such as actual load, temperature, and other environmental factors to predict future scenarios.
(iii) Diagnosing damage caused to the vehicle.
(iv)  Providing a platform to study the effects of modified parameters that were not considered during the design phase.

Any damage detected by DT could be resolved by activating in situ repairs or recommending appropriate changes to the mission, resulting in a longer lifespan and a higher success rate of the mission [45]. By reducing the uncertainty of prognosis and maintenance intervals, DT improves the planning and reliability of future missions [46]. Besides securing the spacecraft and the mission, DT was also used to ensure the safety of the crew members operating in the remote and uncharted territories by testing various possible recovery scenarios in case of emergency [44].

Aside from performance and reliability, one of the benefits of using DT is that it can be used for predicting errors, which makes the maintenance of the spacecraft or aircraft easier and cheaper than the traditional scheduled maintenance. The U.S. Air Force Research Laboratory (AFRL) in 2020 announced that they would be creating a DT of one of their supersonic bombers, B-1B, for predictive maintenance, which will be done by 3D scanning each and every part of the aircraft down to the nuts and bolts [47,48]. Through the scanning process, any structural faults or damages will be discovered which will help in creating a medical record of it. Then data from the aircraft will predict areas that are more likely to have structural issues. Over the aircraft's life cycle, layers of maintenance data, test/inspection results, and analysis tools will be superimposed with the digital model. AFRL has previously signed a contract with Northrop Grumman worth USD 20 million for the development of a DT to forecast the problems with the airframe of different kinds of Air Force aircrafts so that preventive maintenance can be performed [49]. Their experts are trying to improve predictive maintenance for the aircraft by improving flight load data, developing more realistic structural analysis models, and quantifying and reducing modeling uncertainty. An aircraft's life is highly dependent on the load it carries; the life of an aircraft can improve up to 200% with the reduction of just 20% weight [50]. With the help of DT technology, the load spectrum of an aircraft can be recorded and processed for determining its useful life.

A DT for fan-blade reconditioning has been proposed by J. Oyekan et al. [51] which incorporates the use of cameras and vision-based algorithms, along with a robotic arm for grinding the fan blade of the aircraft engine to the required specification in order to improve aerospace maintenance, repair, and overhaul services. S. Liu et al. [52] proposed a DT for the machining processes of an aircraft's rudder which can mimic physical machining processes from three aspects: geometry, behavior, and context. NASA has even developed the DT of the rocket engine to predict real flight conditions and their impacts better on the engine start-up [53].

DT is the enabling aerospace sector to develop new products faster and cheaper while improving the production line quality and reducing the aircraft maintenance time [54]. The new player in the aerospace industry, SpaceX, is also leveraging the DT technology for product designing and testing to address the problems and achieve the required performance before they even build the products, which enables them to develop cheaper rockets [21,55].

The adoption of a DT in the aviation/aerospace industry is highly beneficial, though it can be a slow process due to the regulations and data collection mechanisms from the aircraft, which makes it expensive, as well as having certification concerns for onboard equipment and software [25].

## 3. Manufacturing

Although the development of DT started from the Aerospace industry, the industry which is exploring the technology the most is the manufacturing industry. DTs have been described as the key enablers of Industry 4.0 and Smart Manufacturing [56,57]. Any manufactured product goes through four main phases throughout its life cycle: design, manufacture, operation, and disposal (Figure 3). Smart manufacturers can leverage DTs in all four phases of the product [15,58].

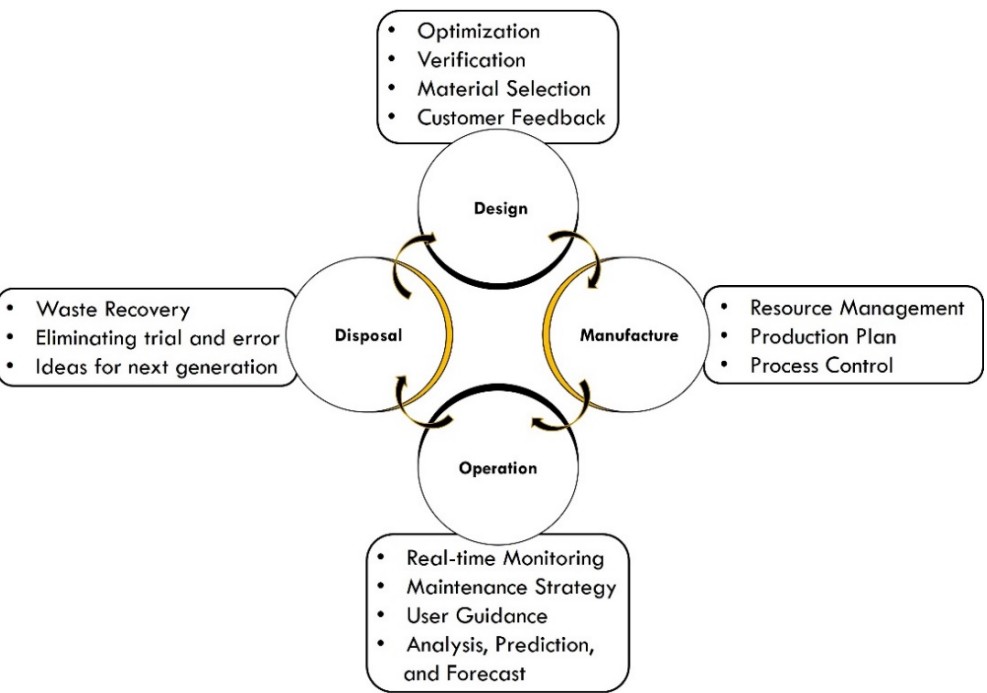

**Figure 3.** DT's applications throughout a product's lifecycle.

During the design phase, DT allows the designers to verify their product design virtually, which enables them to test different iterations of the product and choose the best one [56]. Using the real-time data from the products of previous generations, designers get an insight into the features that are working best for the consumers and those which need improvements [21]. This makes the whole process of improving the design easier and faster. An example of this is the car manufacturer Maserati which used DT for optimizing car body aerodynamics using wind tunnel tests virtually, which are elaborate and expensive otherwise [59]. They also optimized the acoustics of the car inside by using the data from a dummy equipped with microphones in the prototype car. Fei Tao et al. [60], taking a bicycle as an example, have proposed a DT-driven product design framework that can be useful for manufacturers to develop DT to aid the product design process. Unlike traditional bike designing methods which are based on the designers' knowledge and experience, the DT of a bicycle constantly collects the data from the physical space, which could be compared, analyzed, and used for designing or redesigning next-generation bicycles. With customer reviews and usage habits, designers can get a better understanding of customer requirements which can be translated into better and improved features. Capturing customer preferences via a DT lets businesses know about the market trends which can be integrated with the customer usage data to see the effects on product performance. This allows businesses to take design-decisions and incorporate them in an informed way, thus

making the process of integrating the customer feedback into the product to deliver customized products easier [61,62]. Moreover, a DT can help designers in the selection of the material for their products. F. Xiang et al. [63] proposed a DT-driven method for the optimal selection of green material and applied it to laptop shells. From designing to waste management, different manufacturing phases were simulated for different materials and, based on their physical properties, cost, and environmental impact, materials were scored to optimize the selection process of materials.

Turning raw materials into a finished product is the next step in manufacturing. A DT can be a useful tool at this stage in resource management, production planning, and process control [64]. A DT can help in (i) production planning and control by planning and executing the orders automatically and improving decision support by means of a detailed diagnosis, (ii) maintenance by evaluating and analyzing machine conditions, identifying any changes in the production system and its effects, and taking anticipatory maintenance, and (iii) layout planning by evaluating the production system continuously [11,65]. By predicting failure, the DT cuts down the downtime as it allows to either schedule maintenance or take preventive measures. K. Vijayakumar [66] et al., in their research, proposed a method to implement DT for a manufacturing plant, which resulted in a significant time reduction in monitoring the machine states, forklift location, asset location, and constraint management process.

After the sale of the product, when it is in operation, manufacturers can receive the real-time product operation state via its DT and can develop a maintenance strategy accordingly. According to Fei Tao et al. [64], a DT can provide nine types of product services which includes the service of (i) real-time state monitoring, (ii) energy consumption analysis and forecast, (iii) user management and behavior analysis, (iv) user operation guides, (v) intelligent optimization and updates, (vi) product failure analysis and prediction, (vii) product maintenance strategy, (viii) product virtual maintenance, and (ix) product virtual operation.

The final phase of the product, i.e., disposal, is often neglected. Thus, the information which could improve the next generation product/system is often lost when the product is retired [67]. Xi Vincent Wang and Lihui Wang [68] have proposed a novel DT-based system for the recovery of waste electrical and electronic equipment to support the manufacturing and remanufacturing operations throughout the product's life cycle, from design to recovery. The product information models developed by them, from design to remanufacturing, were based on international standards.

In addition to improving the manufacturing process through different phases of a product, a DT also offers benefits to the field of additive manufacturing (AM) [69,70]. Creating a DT of a 3D printing machine can help us in achieving a product with desired properties by eliminating the need to do several iterations of trial and error tests which will, in turn, shorten the time between design and production, thus making the whole AM process time- and cost-effective [71]. The proposed model of DT for the 3D printing machine by T. Mukherjee and T. DebRoy [71] consists of a mechanistic model, a sensing and control model, a statistical model, big data, and machine learning. A. Gaikwad et al. [72] used simulation, in situ sensing, and machine learning to create a DT of the AM process which could detect defects in AM parts during the process. Their DT model combined real-time data from in situ sensors with the physics-based model, along with a machine learning framework to predict flaws with higher statistical fidelity as compared to theoretical model-derived predictions or in situ sensor data alone. DT can also help in making the whole manufacturing process more sustainable and intelligent by making the whole process more autonomous and self-optimized, along with a network of manufacturing equipment, systems, and services that support each other [17,73].

One of the most developed directions in manufacturing, especially in the scope of Industry 4.0 is the robotization and automatization of the production lines [74]. DT is playing a crucial role in this integration as industrial robots are being programmed with mainly three methods, which are closely related to the twinning of the manufacturing equipment.

Those three methods are: (i) Offline, a dedicated virtual environment (usually each robot brand have their own) for programming each aspect of the robotic cell for later deployment through the network to the physical robot; (ii) Online, which is being adapted by means of sensor information, usually being twinned in the dedicated virtual environment (e.g., Robot Operation System (ROS)), and is able to directly affect the pre-programmed path and routine of the robotic systems; (iii) Manual, which is robot programming by the usage of a flex pendant, but with the introduction of Virtual Reality (VR) and Augmented Reality (AR) interfaces, it also uses a twin for manipulation near the virtual robot remotely [75]. As it is seen, combinations of those methods exploit the virtual twin of the robotic cell and are widely used across the various industrial sectors. Moreover, DT is used as a validation tool for human–robot collaboration (HRC) safety standards, to evaluate the safety level of the system first, for example, using VR human avatars, before experiments with real operators in the actual system [76].

DTs in the manufacturing industry are beneficial to future products even before the actual product exists. It helps in designing, testing, process optimization, predictive maintenance, product services, and proper disposal. By detecting early design flaws, DT-based Smart Manufacturing systems can reduce the time and cost of physical commissioning/reconfiguration [77]. In addition, DT helps in a better visualization, which helps in better learning and decision making, as well as in collaboration by giving access to a broader set of professionals to work together. In this way, DTs yield a better understanding of equipment/machinery and processes by all, which leads to better-designed products, and more efficient, time- and resource-saving processes, while also making sure that there is a strong and fluid connection between the various phases of manufacturing [78].

## 4. Healthcare and Medicine

Healthcare is one of the largest industries in the world and its integration in everyday living is ubiquitous. The DT technology in the healthcare system looks at the hospitals, operational strategies, capacities, staffing, and care models to optimize the care, cost, and performance of the healthcare industry, as well as help in informed and strategic decision making [79]. Besides facilities and infrastructure improvements, it also has applications in mimicking patients' behavior, and thus providing them a customized cure and care.

Siemens has developed a DT for the radiology department for a hospital in Dublin, Ireland [80], which was facing challenges with delivering efficient patient care and experience due to rising patient demand, escalating clinical complexity, ageing infrastructure, a lack of space, as well as a longer waiting time, interruptions, and delays. Siemens, along with the hospital, redesigned the layout of the department and tested new operational scenarios by building a 3D computer model of the radiology department and its operations and then applying process optimization using 'Workflow Simulation'. To get a better insight into the department, they conducted a weeklong on-site assessment, which included workshops, interviews of stakeholders, and process observation. Within a few weeks of the implementation of the DT, a significant improvement was seen, resulting in shorter waiting times, a faster patient turnaround, better equipment utilization, and lower staffing costs [81].

In recent years there has been a shift within the industry from 'one size fits all' treatments/medicines towards the 'tailor-made' treatments, as the same disease can develop and affect different people differently based on their socioeconomic and geographic background, genetic make-up, pharmacogenomics, biological gender, age group, family medical history, and lifestyle choices; this is also referred to as personalized/precision medicine. DT technology fits right in the area of precision medicine as the DT outcomes depend entirely on the data fed to it (Figure 4). For personalized treatments, a DT can be conceived for the whole human body, or for just one body system (digestive system, respiratory system, etc.), for a single body organ (liver, heart, etc.), for finer body component levels (cellular, subcellular, or molecular levels), for a specific disease or disorder, or for other relevant organisms (e.g., a virus, interacting with one DT of a human body/organ) [82]. Despite this, unlike other industries, it is hard to establish a connection between humans and their

DT because humans cannot be embedded with sensors. However, once the DT of humans is developed, it will be able to suggest the right treatment for the patient based on its data [83].

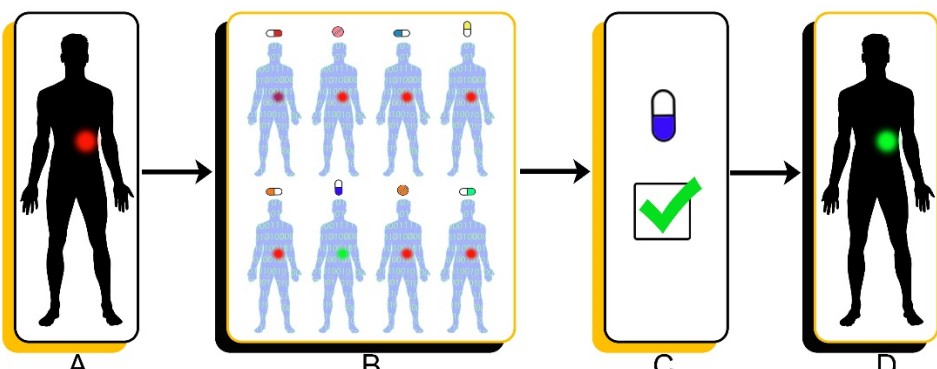

**Figure 4.** DT concept for personalized medicine. (**A**) A patient with a local sign of disease (red). (**B**) Patient's DT is created and virtually treated with different treatments/medicines in various combinations. (**C**) Treatment showing the best results is chosen. (**D**) Personalized medicine is given to the patient to treat them (green). (Adapted from [83]).

Even though human DT seem like a futuristic dream, digital models of organs and body parts have been developed. The Hewlett Packard Enterprise (HPE) launched the 'Blue Brain Project' to reconstruct and simulate a digital model of brains in order to understand the complexities and brain disorders at different levels [84]. Philips has developed 'HeartModel', which combines the data from the patient and a generic model of the heart to create a personalized 3D model of the heart for the early detection and prediction of cardiovascular diseases [85]. They have also developed a software suite called HeartNavigator, designed to assist with pre-surgery plans and tool selection by merging the Computerized tomography (CT) scan images with heart model and X-ray data. Another company, which is working on developing a DT of the human heart, is Siemens. They have been successful in simulating the physiological processes of a heart using magnetic resonance (MR) images and electrocardiograms (ECG) measurements which were used by cardiologists at the University of Heidelberg to predict the results of the treatment options to select the one with the most chances of success [86]. SIMULIA Living Heart by Dassault Systems is also another example of a high-fidelity model of the human heart which turns a 2D scan of the heart into a 3D model [87].

It is not just the big companies working to develop DTs of organs, researchers exploring the field such as R. Velazquez et al. [88] introduced a DT architecture for detecting Ischemic Heart Diseases (IHD) by collecting the patients' data from internal and external sensors, medical records, and social networks, and helping them accordingly; they called it the 'Cardio Twin'. They were successful in implementing it by training their model using ECG data and testing its accuracy with pre-prepared datasets.

The FDA (U.S. Food and Drug Administration) authorities are also considering DTs as a potential tool for clinical trials to accelerate medical innovations along with the regulatory approvals [89]. They collected data from a number of historical Alzheimer's disease clinical trials and used it to train a machine learning model to generate a DT to predict disease progression, and found out that the DTs were statistically indistinguishable from the actual control arm subjects [90]. DT technology is emerging as an exciting tool in the healthcare industry that looks promising for the future. On the one hand, DTs of healthcare infrastructures, such as hospitals, operations, and staff, can optimize patient care, cost, and performance, and on the other, DTs of patients can be a step towards personalized treatment and medicines. Both types of applications of DTs in the healthcare industry will enhance the patients' experience and success of the treatments. However, there are some concerns over certification and regulatory issues, as well as security and privacy when it comes to the implementation of the technology, especially in healthcare [91].

## 5. Power Generation/Energy

Companies involved in the generation and/or sale of energy from either non-renewable resources, such as oil, gas, and nuclear, or renewable resources, such as wind, solar, and hydro, come under the energy sector [92]. The DT technology in the energy sector is being used from wind farms to nuclear plants. General Electronics (GE) has been successful in creating the DT of an entire wind farm. The DT of a wind turbine increases its energy production, optimizes the strategies for its maintenance, as well as improves its reliability by collecting real-time data such as weather, service, and performance reports, among others. GE provides comprehensive hardware and software solutions to its customers for creating, enhancing, and optimizing their wind farm using their cloud-based software DT platform called Predix [93]. Similarly, DNV GL, a company delivering testing, certification and advisory services to the power and renewable industry, developed a DT analytics tool called WindGEMINI for wind turbines which helps to improve the performance of the wind farm by providing predictive maintenance. Two such cases of predictive maintenance are (1) detection of a crack on the blade of the turbine in 2019, which prevented the loss of USD 5000 [94], and (2) in 2017, a defective component in the gearbox affecting 6% in performance was detected, saving GBP 6000 [95]. Apart from this, it also provides an evaluation of energy production in the long term which includes analysis of the power curve, component failures, and the remaining life. Using WindGEMINI, it was found out that even a small reduction of 2% in the power curve performance can save USD 80,000 over the lifetime of a turbine [96]. Furthermore, the DT of wind turbines can be used in designing and testing them, as well as finding the best position/location to maximize energy generation by taking factors into account such as wind speed, waves, temperature, etc. [97].

The predictive and interrogative features of DTs are being exploited for the next generation of power grids. Siemens developed DTs for the planning, operation, and maintenance of the Finnish power grid by creating a single digital grid. Its advantages, besides improved safety and reliability, include: (i) resource-saving by transforming most of the manual work into the automated one for simulations, (ii) the efficient and improved use of data, (iii) the easier use of data for new applications, and (iv) the improved decision making using big data [98]. M. Zhou et al. [99] implemented and tested their DT on a large-scale grid network model exported from the Chinese national grid energy management system for real-time monitoring. They were successful in tracking the operation state of the power grid with only a sub-second delay. Following Finland, Norway is planning to utilize a DT for optimizing the operations of the power network with the new complexities brought by intermittent renewables and distributed energy resources [100]. The DT will allow operators to predict grid conditions, balance grids, and prevent blackouts.

DT technology, when applied to nuclear power plants (NPP), can improve the algorithms for controlling and testing the plant, the diagnosis of plant equipment such as pumps, motor, valves, etc., and the parameter tuning of automatic regulators [101]. France has announced a four-year project led by Électricité de France (EDF) to develop DTs for the country's nuclear reactors [102]. EDF operates a fleet of 56 power reactors in France, which is why, instead of creating a DT of a generic rector, EDF focuses on integrating data specific to each plant facility, which includes design data, operating records, and real-time measurements. One of the main applications of these DTs will be training new operators and engineering students. Unlike the real world, the DTs of these nuclear plants will make it possible to fully immerse in a functioning reactor virtually and access all kinds of information on the behavior of its components. A multidisciplinary team, consisting of the faculty from the University of Michigan, senior scientists from the Idaho National Laboratory and Argonne National Laboratory, and industry partners from Kairos Power and Curtiss-Wright, are also working on developing DTs for nuclear reactors [103]. Their focus is to use these DTs for predictive maintenance by using machine learning algorithms so that some expenses in constructing new nuclear plants could be saved, unexpected outages could be avoided, and maintenance could be optimized. DTs of NPPs are not limited to maintenance and training. By collecting, storing, and managing the data, the DT

can be effectively used at all stages of the life-cycle of the NPP units, so that no valuable information is lost after the decommissioning of the plant, and the same information can be used to design the next generation of plants [104].

As the demand for renewable energy sources increases, so does the demand for their DTs which can be used for their optimal design, maintenance, and performance, as well as for energy distribution. According to Siemens, the implementation of DT is a prerequisite if we want to save the planet from the detrimental effects of climate change [105]. The DT can play a crucial role in optimizing the production and distribution of electricity, as well as in the maintenance of the assets which are involved in the generation of energy.

## 6. Automotive

As in other industries, predictive maintenance is one of the applications of a DT in the automotive industry. Rajesh P.K et al. [106] show the role of DT in predictive maintenance, taking the brake pad as an example. They collected and compared the real-time data from a brake pad and the data from the simulated DT of the brake pad. The similarity in the data from both sources suggests that the predictive maintenance of brake pads is possible using their model and collecting the real-time data for a longer period of time. The automotive industry is also exploiting DT technology in order to provide more personalized/customized services to its customers by capturing and analyzing behavioral and operational data of the vehicle [107]. By doing so, car manufacturers can build a car that fits the needs of their customers by keeping track of the features depending on their usage. Tata Consultancy Services (TCS) believes that, besides doing monitoring and predictive maintenance for vehicle services and parts, a DT can boost vehicle sales as well [108]. A DT can provide a 360° view of the vehicle, integrating customer's preferences which, when combined with AR, can alleviate the whole sales experience by making it more immersive and interactive.

Tesla Motors is developing a DT for every car it manufactures. By using the data from individual vehicles, the company makes sure that every one of its cars is performing as intended. With the data, Tesla also updates the software for each car individually and then uploads it to fix several maintenance issues, such as compensating a rattling door by adjusting hydraulics [109,110]. Another automobile company that is using DTs is Volkswagen. Unlike Tesla, Volkswagen used DTs to integrate a new robot workstation in one of their manufacturing plants. They created a highly detailed 3D model of their plant which included robot arms, sensor logic, and safety components, and then simulated processes and procedures before adding the production line. The DT saved them about three weeks of time and 40 square meters of production space [111]. Maserati also used a DT to add two new assembly lines into an existing facility. They saw how the changes in the car design affect production using a DT and then adapted the production processes accordingly [59].

A DT has even found its way into competitive sports such as Formula 1 (F1) racing. The team of Mercedes-AMG Petronas Motorsport has developed a Digital Twin to analyze their car performance after every race. A total of 150 sensors in the car collect data of temperature, pressure, acceleration, forces, shaft speed, etc., every 0.001 s, to which virtual sensing is added, thereby having five billion data sets to be processed by the end of a two-hour race [112]. With the help of a DT, race engineers study the data to build a car that can perform better, as well as having increased reliability and safety. All the data from the race help the team in developing strategies for future races. Moreover, the F1 drivers now use the simulations to prepare for the real races, test new design features, and gain a deeper understanding of the vehicle's behavior [113].

Another area in the automotive sector where DTs are finding their applications is the autonomous vehicles sector. Each autonomous vehicle is validated for the safety and movement algorithms aspects in the commercial or open-source digital environments. As most of the vehicles are unique in their development and manufacturing process, manufacturers are working on twinning every single aspect of the vehicles, such as electric

propulsion drives, which is common for different brands and types of machines. Connected with a physical drive test bench, the simulated vehicle can drive around the virtual world based on the physical data and transfer back the behavior of the machine, which makes the physical drive behave as it should if it was installed in the car [114,115].

All the applications of DTs described in manufacturing can be applied to this industry from design to disposal. Thus, DTs in the automotive industry can be beneficial for manufacturers in their manufacturing process, as well as the car dealers in selling them, by providing a more interactive and immersive customer experience. Not only that, but it can also give the opportunity to the customer to customize their vehicles as per their desire. DTs are bringing value to every user, and it does not matter if it is a normal driver or a competitive one.

## 7. Oil and Gas

The oil and gas sector generates about USD 3.3 trillion annually in revenue, making it one of the largest sectors in the world in terms of dollar value [116]. The emergence of DT technology in the oil and gas industry has brought tremendous value to this industry by being an effective tool in, (i) reducing and a better understanding of risks, (ii) creating and managing executable work schedules, and (iii) identifying any changes in the process and responding accordingly [117]. Additionally, DT can be a critical resource for oil companies when it comes to extracting offshore resources [118]. Due to all these benefits of DT, Accenturem, in one of their surveys from 2017, found that out of 200 refineries, 66% of them were planning to increase their investments in digital transformation in the coming years, and 57% already had invested more than the previous year [119].

BP (British Petroleum) [120], one of the largest oil and gas companies in the world, has been using DTs for several of their plants in the North Sea called APEX. APEX helps them in safely optimizing the production, remote surveillance, predictive maintenance, saving time in optimizing, and testing. The most notable benefit contributed by DTs for BP was a boost in oil and gas production by 30,000 additional barrels per day globally [120].

Another oil and gas company to embrace the DT technology is Royal Dutch Shell. By collecting more than 10 million operational data per minute, they use it for maintenance, improvement in productivity and safety, and reduction in emissions by anticipating the conditions to optimize asset performance and managing them autonomously [121]. They also developed an improved fatigue model for their assets using the DT concept by combining the data from sensors with a structural finite element model in order to accurately predict the life of assets so that they can optimize the inspection planning and safety cases around it [122]. Other companies, namely Eni [123] and Equinor [124], are also exploiting the technology for maximizing productivity while keeping the costs and risks at a minimum.

Siemens [125] also studied the benefits of deploying DTs in the oil and gas industry and found out that the project cycle was shortened by four–eighteen weeks in total: four weeks in stabilizing the operation and eight weeks in the engineering phase. Moreover, the capital expenditure and operation expenditure were also reduced by USD 4–7 million and USD 60–100 million, respectively. It was also suggested that creating a DT as a part of the project rather than retrofitting it to the existing plant is more economical.

Since the oil and gas industry relies on heavy and sophisticated plants as well as machinery located in remote areas under extreme environmental conditions, a DT makes it safer to monitor and control the activities/processes, thus reducing the risks involved. Moreover, by predicting the downtime, the DT improves the overall process, which directly translates into savings both in terms of time and money.

## 8. Smart City

A smart city is the one that utilizes information and communication technology to efficiently run it, by sensing, analyzing, and integrating the key information of core systems and thus making intelligent decisions to the needs of the city, which include livelihood, environment, public safety, city services, industrial/commercial activities, etc., [126]. In

smart cities, DTs have found their application in urban planning also, to improve the quality of life of people. One such example is Carson City in Nevada, USA. Carson City Public Works Department developed a DT for the management of their water supply, which resulted in supplying water to 50,000 citizens of the city efficiently and a 15% reduction in operational hours [127]. Using a DT for planning and decision making can lead to cities that are more economically, environmentally, and socially sustainable.

In 2018, the National Research Foundation (NRF) of Singapore created a 'Virtual Singapore', combining 3D maps, city models, and a data platform with fine details including texture, building materials, geometry, and components of facilities, etc. NRF believes that Virtual Singapore will be beneficial not only to the government, but also to its citizens, business owners, and other research communities by acting as a testbed for new ideas and by providing information that will help in the decision-making process, as well as in resource planning and management. It can also be used to improve the accessibility of a specific area and simulate emergency situations [128].

Another city adding to this list is Amaravati, India. The DT of the city is being designed for a population of 3.5 million people costing USD 6.5 billion. The project is going to cover an area of 217 km$^2$, which includes 134 km of the metro network, 316 km of main roads, over 100 hospitals and schools, 40 colleges, 3 universities, and 3 state government buildings [129]. According to Cityzenith, the company responsible for developing the DT of the city, Amaravati will be one of the top three digitally advanced cities in the world on completion. Having cities on digital platforms will allow their stakeholders to monitor the construction progress in real-time, the environment and overall wellness of the city, mobility and traffic, the microclimate and climate change, etc. Moreover, it will serve as a portal to all citizens for all government information, notifications, forms, and applications [130]. A similar project has been launched by the government of Victoria, Australia, to create a DT of a 480 hectares area called Fishermens Bend near Melbourne, Australia. The project will have real-time data visualization of public transport and building occupancy, tools for planning analysis and analytics, traffic flow predictions, power and water usage forecasts, etc., [131].

The DT of a smart city can be used for disaster management or emergency response plans [132,133]. The DT of the Docklands area in Dublin, Ireland, developed by G. White et al. [134], was fed with the rainfall amount and river levels data to predict when flooding could occur, which could be used to alert the citizens of possible floods. In smart cities where flooding has been identified as a problem, historical data of the city can be used for creating a longer-term flood prevention mechanism, planning city skyline, or for large-scale projects such as introducing water storage areas or diverting rivers accordingly.

A DT can change the way we look at our cities and living spaces. It can help authorities in resource allocation, urban planning, and sustainable development. It can also give urban designers as well as architects, engineers, constructors, property owners, and citizens an opportunity to study and analyze the city's infrastructure in different scenarios and assess any future risks, thus improving the overall performance of a city, its infrastructure, processes, and services [135]. With the participation of every stakeholder of the city, from citizens to the government cities via the DT, cities can become more democratic as everyone can have an opinion and say on what is lacking in their communities and how can their environment and city services can be improved [136].

## 9. Mining

The mining sector, which is dedicated to locating and extracting metal and mineral reserves from the earth's surface, is considered one of the oldest established industries [137]. Just as for the oil and gas industry, a DT is a perfect tool for process/operation optimization on mining sites. Ernst and Young has identified a DT as one of the key enabling technologies in the mining industry as it is going through digitalization [138]. In their report, they presented four areas where DTs contribute to mining: (i) Mining Operations: predictive maintenance improves asset reliability and cuts down unplanned downtime, (ii) Processing: optimizing plant set points improve the efficiency of processes and the

quality of the product, and reduces bottlenecking, (iii) Transport: predicting and optimizing the transport network improves transportation reliability, and (iv) End-to-End: simulating and analyzing different scenarios across the value chain determines optimal plans and schedules. Moreover, DTs help miners in testing new processes or machinery before using the real one on the site and also trains new miners [139], especially for emergency situations, by educating them about the correct procedure and training them remotely in a risk-free and stress-free environment [140].

PETRA DataScience [141], a provider of digital mine-to-mill software for value chain optimization, created the world's first DT for the optimization of the mine value chain. Their machine learning DT uses two years of historical data to simulate 'mine planning, blasting, metallurgy, and process control options.' The Ban Houayxai Gold–Silver Mine in Laos used the DT, which, using the data on the types of rock, optimized the crushing efficiency during drilling and blasting, thus increasing the recovery rate of the metal from the mine [142].

Anglo American plc [143], a multinational mining company, is using the DT in their mining sites in Chile and Brazil to reduce the fuel consumption of their trucks, thus optimizing the mining fleet. By tracking the performance of the haulage fleet, they can analyze the data which is used for improving equipment efficiency, condition monitoring, and predictive maintenance. They are also planning to implement DT for their pipelines, smelters, and refineries to make the process more effective and efficient. Rio Tinto [144], the world's second-largest metals and mining corporation, also uses DT in the Gudai-Darri (Koodaideri) mine. Using a DT, they are safely testing new ways to improve their productivity without damaging any parts or halting any process within the plant.

DT technology has the potential of adding new value to the mining industry, from increasing efficiency to providing a safer environment, which explains why mining corporations are investing in DT technology. The International Data Corporation (IDC), a global provider of market intelligence and advisory services, has predicted that 70% of the mining companies will be investing in DT technology in the next two years [145]. In addition, DT has caught national interest for the 'revitalization of domestic resource development' [146].

## 10. Maritime and Shipping

Anything or any activity related to the ocean, sea, ships, navigation of ships from one destination to another, seafarers, etc., is represented by the maritime industry which also includes the transportation of passengers and freight on both inland and international waters [147,148]. Globally, around 80% of trade by volume and 70% by value is carried by sea, according to the United Nations (UN) [149], making it the most important means of transport for goods. Similar to the aviation industry, the main application of DT in this industry focuses on increasing the reliability of the assets, improving maintenance, and reducing operational costs.

Military Sealift Command, the leading organization for providing military transport ships to the United States Navy and Department of Defense, with the help of GE, is building a DT for their cargo ammunition vessels [150]. The data collected from marine equipment, such as variable frequency drives, propulsion motors, diesel engines, and generators, are used to check the real-time performance of the vessel against the calculated one, and any performance deviations which indicate potential failure in the engines or other critical infrastructure are reported and resolved before it ever happens, thus improving their assets' and missions' availability, efficiency, operations, and readiness. The DT also makes remote monitoring and diagnostics possible [150].

In 2019, DNV GL, the creator of the aforementioned WindGEMINI, which is also the world's leading classification society and a recognized advisor for the maritime industry, implemented DT technology on one of the world's largest crane vessels. The vessel demands constant inspections due to its large lifting capability of 14,000 tons, which puts the vessel under high stress. The DT makes reporting, assessing problems, maintenance planning, and predictive analysis easier, which results in huge cost savings [151]. DNV GL

believes that DT can bring significant benefits to everyone involved in the maritime industry, including ship owners, equipment manufacturers, authorities, universities, maritime academies, and consultancy services [152].

The DT technology is not restricted to just vessels/ships, but it is finding its way to ports as well. The Port of Rotterdam [153] has sensors throughout its docks that gather real-time data of the environment around and water conditions, which includes air temperature, wind speed, humidity, turbidity, water salinity, flow, levels, tides, and currents. The port even has 'Digital Dolphins', smart quay walls, and sensor-equipped buoys. In addition, they have a physical container, Container 42, fitted with a sensor, which uses AI to optimize the time for the vessel to sail and several cargos to carry. They are planning to become the first digital port by 2030.

## 11. Agriculture

Enterprises in the agriculture sector are involved in harvesting and producing agricultural commodities such as crops, livestock, poultry, fish, as well as fertilizers, packaged foods, and agricultural machinery, etc. [154]. Due to the growing human population, the agriculture industry is under the extreme burden of increasing its production to meet the nourishment demands of this record number of human beings. According to the Food and Agriculture Organization of the UN, the global population in just 30 years, i.e., in 2050, will reach 9.8 billion, which will create a demand for food, feed, and biofuel 50% higher than it was in 2012 [155]. To achieve this productivity goal, humans will need access to every tool available. Developing a DT for smart farming can enable sustainable development and increase food security for the global population [33]. Although DT technology is still in the early phases of development in the agriculture industry [34], potential applications in this field have been identified.

Verdouw and Kruize [156], in the Asian–Australasian Conference on Precision Agriculture, presented potential applications of DTs in different fields within agriculture, such as animal husbandry, apiculture, crop storage, agriculture equipment/machinery, etc. The DT applications explored by them include:

- Remote monitoring of livestock for the detection and analysis of their health including heat and estrus cycles as wells as the animals' movement tracking.
- Identification of pests or diseases in plants.
- Management and optimization of production plants and replenishment routes by monitoring the stocks of the silos.
- Evaluation of cost-effectiveness of crop management treatments along with tracking machinery in real time.
- Detection and identification of flies in olive farms to use pesticides effectively.
- Monitoring bee colonies for diseases/infections and managing honey storage.

J. Monteiro et al. [157] presented a reference model for the implementation of a DT in vertical farming by enriching the physical system by sensors collecting data on temperature, humidity, luminosity, and the relative $CO_2$ concentration, which is stored on the cloud. Using intelligent data analysis, the system will be able to suggest methods to plan vertical farms and increase production. A similar framework comprising of modeling and analysis, simulation, big data, and visualization for livestock farms was presented by S.K Jo et al. [158]. Besides managing livestock and farms, the DT can promote sustainability. Agriculture uses 11% of the world's land and 70% of fresh water; it is also responsible for 80% of deforestation [155]. DT technology can help by tracking any changes in carbon emission, biodiversity, pollination, and water catchment services, and their causes which will require the research and development of correct sensors, models, and interfaces for these specific purposes [159]. Matthew J. Smith [160] shows concern over the negligence of the real-world systems due to developments of DTs, which can lead to less empathic and intellectually and emotionally distant farmers. The author suggests that farmers and agricultural organizations should make sure to pay attention to the aspects that are not a part of the digital model while taking advantage of the technology to grow.

## 12. Education

Education is an interesting area where DT has started finding its applications, especially in technical courses where the systems are complex and hands-on training on these systems is an integral part. A DT is a great tool for communication in education as it represents multiple domains and visualizes the performance of systems and sub-systems, which results in a better understanding of systems, thus simplifying and accelerating knowledge exchange in a range of technological disciplines [152,161]. DT technology can improve learning and motivate students for studying [162]. The benefits of using DT for learning include [163]:

- Authentic learning experiences which promote effective knowledge construction, skill mastery, learning transfer, and self-efficacy.
- Learning about the physical twin behavior in the real world under different operational conditions.
- Getting immediate feedback on system behavior leading to issue identification and problem-solving.
- Inquiry-based learning during system development and testing.
- Each student can work on an individual DT which is unlike the case where students had to share limited resources.
- DT is a great tool for distant learning students, for whom accessing a physical twin is not possible.
- DT ensures the safety of both students and equipment.

The traditional engineering courses provide their students with specialized knowledge which can be applied to a particular industry which was apt for the traditional industrial structure where departments and divisions work in isolation from each other. As more industries are taking the route of digitalization, the silos between different departments requiring specialists are diminishing, and the need for cross-disciplinary expertise is rising. To address this problem, S. Nikolaev et al. [164] introduced a module in the Skolkovo Institute of Science and Technology, Moscow, Russia, for master's students focused on developing a DT for designing, prototyping, and testing complex systems, which was an unmanned aerial vehicle in their case. Similarly, the University of Southern California has also proposed a pedagogical DT testbed for engineering students, which can enhance their learning experience by giving them an opportunity to explore the complexities and behavior of the physical systems by interacting with their DTs [163]. As the demand for DTs in several industries will accelerate, so will the demand for competent engineers and technicians. In the future, we can expect more courses to be introduced in colleges and universities focused on DTs.

## 13. Construction

The construction industry is not only a very laborious and time-intensive industry, but also an information-intensive industry. From conceptual planning to decommissioning, throughout the project lifecycle of construction project a lot of data is generated [165]. Thus, each step or phase of the project lifecycle requires proper communication and information flow among different stakeholders involved, such as the architects, engineers, contractors, facility manager, and construction workers, which can be achieved using DTs [166]. Just as for the manufacturing sector, DTs in the construction sector can also be used in different lifecycle phases of construction projects: the designing and engineering phase, construction phase, operation and maintenance phase, and demolition and recovery phase [31]. Table 2 lists out all the actual and potential applications of DTs in all phases of any construction project.

**Table 2.** Actual and potential applications of DTs in different phases of construction [31,165].

| Phase | DT Applications |
|---|---|
| **Designing and Engineering** | • Helps in building information modeling.<br>• Reduces overall design process.<br>• Minimizes the possibility of incurring additional costs during rework.<br>• Helps in problem-solving by allowing data to be added, modified, and verified against real-life scenarios.<br>• Allows collected data to be used by designers for future projects.<br>• Improves decision-making regarding feasibility of project, material selection, energy analysis and management, procurement, supplier selection, sustainability issues, etc. |
| **Construction** | • Assesses the structural system integrity.<br>• Ensures building does not fail when forces are applied to it.<br>• Promotes modular construction activities.<br>• Prepares as-built drawings when the design drawings are not available.<br>• Assists in managing resources, materials, schedule, quality, sequence, etc. |
| **Operation and Maintenance** | • Monitors the condition of building assets continuously.<br>• Optimizes maintenance and services in the building<br>• Aids predictive maintenance and ensures well-informed decision making.<br>• Monitors logistics processes<br>• Simulates building energy consumption. |
| **Demolition and Recovery** | • Helps in problem solving by using data from the predecessor for the next generation buildings having similar characteristics.<br>• Conserves and preserves heritage assets virtually that need demolition.<br>• Identifies potential hazards, technical solutions, possible actions to be undertaken regarding the conservation of their assets. |

One of the biggest advantages of using DTs in the construction sector is in building information modeling (BIM) which is a system that generates and manages information on a construction project across its lifecycle. BIM is still mostly used for resource efficiency enhancement and knowledge exchange during the design and construction phase for facilitating the tasks of building stakeholders and avoiding costly design mistakes, which are easier to achieve using DTs [167]. Moreover, DTs promote incorporating sustainability right from the designing phase. Using a DT, the carbon footprint and energy consumption of any building can be estimated upfront, which can be taken into account while designing and building it. S. Kaewunruen and N. Xu conducted a modeling simulation of King's Cross station in London using its DT, and found that the insulation of cavity walls with a fire barrier is the cheapest option for the fire prevention design for the building, which also incurs the lowest carbon emission rate among all options [168].

From the planning to the operating phase, a DT can be a valuable tool in the construction industry. This industry is very dynamic, and the requirement for spaces keeps changing from the original concepts and usage of the building. With a DT it becomes easier to keep track of all the changes and see the effect of these changes. The application of a DT has the potential to transform the construction industry. Therefore, the industry must be opened to change and embrace the opportunities that come with it. Adopting DTs is the only way the construction industry may keep pace with other industries [31].

## 14. Retail

The retail sector includes all the stores and shops that sell the products or goods to the consumer/customer directly. The success of any retail business depends on its customer base; attracting new customers is not enough, and maintaining/sustaining the existing ones by providing the best customer support is equally important. DT technology in the retail sector offers great potential in terms of consumer experience and marketing [35]. By

creating a DT of the customer, retail shops will be able to provide a unique and tailored customer experience based on customers' patterns of interest [169]. Customer satisfaction can be enhanced further by providing them with relevancy-based suggestions based on their DT without irritating them with multiple recommendations [170]. DT technology enables retailers to turn their products into a dynamic platform that continuously acquires data regarding customers' needs and buying behavior in order to provide them with better products and services [171]. Combining the data from a customer's DT with machine learning to understand the customer's behavior will be beneficial for the customers as well as the retailers [172].

Another advantage that the retail industry can take from DT technology is in its logistics and supply chain. DT can be used to track and trace the products within the supply chain in order to have more efficient and reliable inventory planning, aggregate planning, and demand forecasting [173]. A French supermarket chain, Intermarché, created a DT of their brick-and-mortar store user data from IoT-enabled shelves and sales systems to get stocks details in real time and test the efficacy of different store layouts before implementing them [174]. Due to disruptions in the supply chain by the COVID-19 pandemic, 82% of retailers are changing the way it is managed and thus adopting DTs as a tool to address it [175]. DTs provide real-time monitoring, which can enable retailers to implement new non-linear supply chain fulfilment models which are more efficient. Moreover, in the cases of emergency or disruptions, such as the COVID-19 pandemic, DTs can be used to simulate different what-if scenarios in the digital world before making any decisions in the real world. The pioneering companies in developing a DT of their supply chains are BMW and 3M [176,177].

The implementation of DTs in retail is relatively new, as compared to other sectors, but the benefits are plenty, mainly in stores and supply chains. According to Boston Consulting Group, an American management consulting firm, DT has helped retailers in Capex reductions by 10%, sustainable inventory reductions by 5%, and improved EBITDA (earnings before interest, taxes, depreciation, and amortization) by 1–3% [178]. Owing to these benefits, we can expect the adoption of DTs in retail more in the coming years.

## 15. Conclusions

DT technology presents an opportunity to integrate the physical world and virtual world, which can be utilized in addressing the challenges faced by different industries from aeronautics to maritime and from manufacturing to retail. There has been objective progress since the inception of DT technology (Figure 5), however, practical applications and implementations of the technology in the industry remain uncharted territory. Industries such as manufacturing, which pioneer DTs, are leveraging the technology the most by offering improved products by collecting, analyzing, and interpreting data from the DT and retrofitting it to products in the market. The applications of DTs for any product can be realized through its lifecycle, from design to disposal, be it in the manufacturing or in the construction. The benefits of DTs, such as remote monitoring and maintenance, have been realized by industries, especially after the COVID-19 pandemic outbreak in 2020. Companies such as GE have saved USD 1.5 B using the real-time monitoring capabilities of DTs, resulting in lower operation and maintenance costs [179]. Table 3 summarizes the applications and advantages of DTs in each sector. It concisely shows the commonality and contrasting application of DTs in different sectors, along with the benefits it brings to the respective sectors.

In broader terms, the applications in different sectors can be summarized as (Figure 6):

- Optimization
- Decision making
- Remote access
- Training and documentation
- Designing/planning
- Real-time monitoring

- Maintenance
- Safety

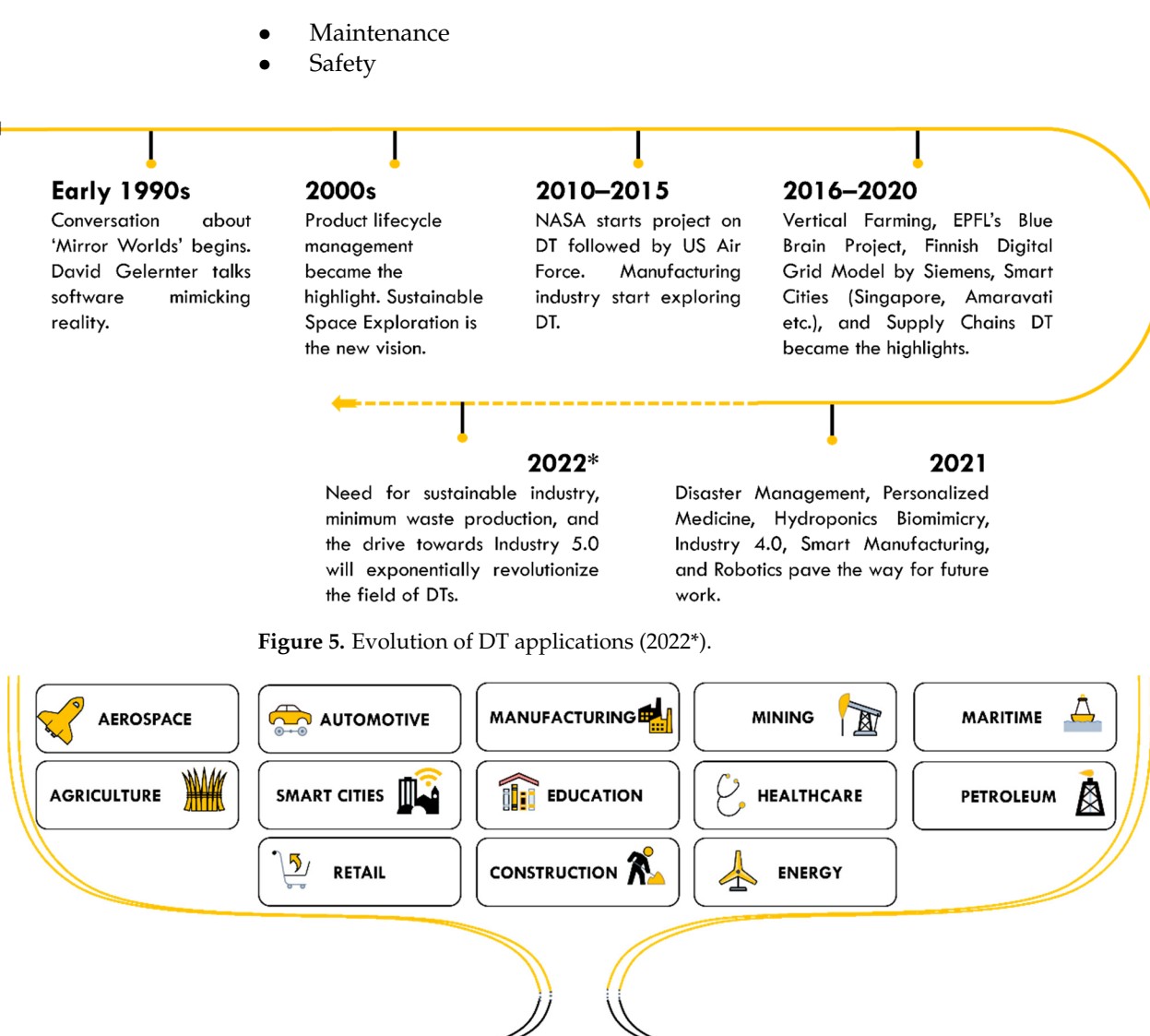

**Early 1990s**
Conversation about 'Mirror Worlds' begins. David Gelernter talks software mimicking reality.

**2000s**
Product lifecycle management became the highlight. Sustainable Space Exploration is the new vision.

**2010–2015**
NASA starts project on DT followed by US Air Force. Manufacturing industry start exploring DT.

**2016–2020**
Vertical Farming, EPFL's Blue Brain Project, Finnish Digital Grid Model by Siemens, Smart Cities (Singapore, Amaravati etc.), and Supply Chains DT became the highlights.

**2022***
Need for sustainable industry, minimum waste production, and the drive towards Industry 5.0 will exponentially revolutionize the field of DTs.

**2021**
Disaster Management, Personalized Medicine, Hydroponics Biomimicry, Industry 4.0, Smart Manufacturing, and Robotics pave the way for future work.

**Figure 5.** Evolution of DT applications (2022*).

**Figure 6.** Applications of DTs in different sectors.

The diverse applications of DTs make them a helpful tool, irrespective of the industry it is being used in. This paper identified 13 industries that are leveraging the technology and reported real-world industrial cases of DT applications that are driving the respective industry forward. Even though the technology has so many advantages, it comes with many challenges too, such as it being a novel technology, its cost and time for implementation, a lack of standards and regulations, and security concerns. These challenges associated with DT technology along with its future have already been discussed in detail in a previous paper [2]. Even though the actual implementation of the DT in these industries is unrealized, speculation of its potential spurs sways with freedom in the imagination of futurists and technologists driving it forward. Therefore, identifying and understanding the potential of DTs in any sector and implementing them for the right application is important as it offers numerous advantages, from simulation and prediction capabilities to record keeping and troubleshooting.

**Table 3.** Applications and advantages of DTs in different sectors/industries.

| Sector | DT Application | Advantages |
|---|---|---|
| **Aerospace and Aeronautics** | • Optimizing performance and reliability of the spacecraft/aircraft<br>• Predicting and resolving maintenance issues<br>• Continuous mirroring of the actual flight to predict future scenarios<br>• Designing and testing of product<br>• Simulating and optimizing product and production systems | • Safer missions<br>• Maximizes the mission success.<br>• Cheaper spacecrafts<br>• Lower operational and maintenance costs |
| **Manufacturing** | • Designing and testing of product<br>• Material selection<br>• Customizing the products<br>• Optimizing production planning and control<br>• Layout/Floor planning<br>• Predicting maintenance issues and developing a maintenance strategy<br>• Real-time monitoring of production and service<br>• Remote troubleshooting of equipment<br>• Analyzing and forecasting energy consumption<br>• Analyzing user behavior<br>• Recovering the waste<br>• Validation tool for HRC safety standards<br>• Collaboration tool | • Better-designed products<br>• Fewer design iterations<br>• Faster and cheaper production<br>• Increases reliability of equipment and production lines<br>• Reduces downtime<br>• Improves decision support<br>• Lower maintenance costs<br>• Less wastage |
| **Healthcare** | • Optimizing the care, cost, and performance of hospitals, operations, staff<br>• Informed and strategic decision making<br>• Personalized cure and care<br>• Detecting and diagnosing disease | • Efficient patient care<br>• Improves success of the treatments<br>• Shorter waiting times<br>• Faster patient turnaround<br>• Better equipment utilization<br>• Lower staffing costs |
| **Energy** | • Finding the best position/location for maximum energy generation<br>• Providing predictive maintenance<br>• Optimizing the strategies for its maintenance<br>• Designing and testing<br>• Training new operators and engineering students<br>• Optimizing distribution of electricity | • Increases energy production<br>• Improves assets' safety and reliability<br>• Prevents blackouts<br>• Resource-saving<br>• Improves decision making |
| **Automotive** | • Predictive and remote maintenance<br>• Optimizing production<br>• Floor planning<br>• Providing personalized/customized vehicles and services<br>• Proving immersive and interactive customer experience<br>• Developing strategies for car races<br>• Training professional racers | • Boost vehicle sales<br>• Increased reliability and safety<br>• Improves performance of racing car and racer |
| **Oil and Gas** | • Tool for understanding and reducing risks<br>• Creating and managing executable work schedules<br>• Identifying any changes in the process and responding accordingly<br>• Optimizing the production and asset performance<br>• Remote surveillance<br>• Predictive maintenance<br>• Optimize the inspection planning and safety cases around it | • Lower capital and operation expenditure<br>• Reduces time in optimizing and testing<br>• Boosts production<br>• Improves safety<br>• Reduces emissions<br>• Prevents downtime |

**Table 3.** *Cont.*

| Sector | DT Application | Advantages |
|---|---|---|
| **Smart City** | <ul><li>Resource planning and management</li><li>Testbed for new ideas</li><li>Help in the decision-making process</li><li>Real-time monitoring of construction progress, mobility and traffic, environment, and overall wellness of the city</li><li>Planning emergency response for disaster management</li><li>Studying and analyzing city's infrastructure and assessing any future risks</li><li>Platform to facilitate comments and suggestions by every stakeholder</li></ul> | <ul><li>Improves the accessibility of specific areas</li><li>Promotes stakeholders' participation in decision making</li><li>Improves city services</li><li>Fosters sustainable development</li></ul> |
| **Mining** | <ul><li>Process/operation optimization</li><li>Predictive maintenance of assets</li><li>Predicting and optimizing transport network</li><li>Simulating and analyzing different scenarios across value chain</li><li>Trains new miners, especially for emergency situations</li></ul> | <ul><li>Improves asset and transportation reliability</li><li>Cuts down unplanned downtime</li><li>Improves processes' efficiency and product quality</li><li>Risk-free and stress-free environment</li></ul> |
| **Maritime** | <ul><li>Predictive maintenance of vessel</li><li>Remote monitoring and diagnostics</li><li>Maintenance planning</li><li>Optimize the time for the vessel to sail and number cargos to carry</li><li>Tool for digitalization of ports, quay walls, buoys</li></ul> | <ul><li>Improves availability, and efficiency assets and missions</li><li>Increases reliability assets</li><li>Improves maintenance</li><li>Reduces operational costs</li></ul> |
| **Agriculture** | <ul><li>Remote monitoring of livestock</li><li>Identifying pests or diseases</li><li>Managing and optimizing production plants</li><li>Monitoring the stocks of the silos</li><li>Evaluating cost-effectiveness of crop management treatments</li><li>Planning for vertical farms</li></ul> | <ul><li>Promotes sustainable development</li><li>Increases production and thus food security</li><li>Better decision making</li></ul> |
| **Education** | <ul><li>Tool for communication for visualizing working of systems and sub-systems</li><li>Teaching physical twin behavior in the real world under different operational conditions.</li><li>Issue identification and problem-solving by immediate feedback on system behavior</li><li>Inquiry-based learning during system development and testing</li><li>Sharing unlimited resources amongst students</li><li>Tool for distant learning students</li></ul> | <ul><li>Better understanding of systems</li><li>Simplifies and accelerates knowledge exchange</li><li>Ensures safety of students and equipment</li><li>Opportunity to explore the complexities and behavior of the physical systems</li></ul> |
| **Construction** | <ul><li>Helps in BIM</li><li>Problem-solving against real-life scenarios.</li><li>Tool for future projects.</li><li>Decision-making regarding feasibility of project, material selection, energy analysis and management, procurement, supplier selection, sustainability issues, etc.</li><li>Monitoring and assessing condition and integrity of building continuously</li><li>Optimizing maintenance and services in the building</li><li>Monitors logistics processes</li><li>Conserving heritage assets virtually that need demolition</li><li>Resource efficiency enhancement and knowledge exchange</li></ul> | <ul><li>Reduces overall design process and associated cost</li><li>Promotes modular construction activities</li><li>Avoids costly design mistakes</li><li>Promotes incorporating sustainability</li><li>Easier to keep track of all the changes and its effect</li></ul> |

**Table 3.** *Cont.*

| Sector | DT Application | Advantages |
|---|---|---|
| **Retail** | <ul><li>Providing tailored customer experience and suggestions</li><li>Optimizing logistics and supply chain</li><li>Inventory and aggregate planning</li><li>Demand forecasting</li><li>Real-time monitoring to implement new non-linear supply chain fulfilment models</li><li>Store layout planning</li><li>Decision making in the cases of emergency or disruptions such as COVID-19</li></ul> | <ul><li>Better products and services</li><li>Improves sales</li><li>Sustainable inventory reductions</li><li>Decreased operational cost</li></ul> |

**Author Contributions:** Conceptualization, M.S. and E.F.; writing—original draft preparation, M.S.; writing—review and editing, R.S., E.F. and V.K.; writing—review, Y.Q. and D.D.; supervision, R.S., E.F., D.D. and N.M. All authors have read and agreed to the published version of the manuscript.

**Funding:** This research was funded by grant number 16/RC/3918 and "The APC was funded by 692 Science Foundation Ireland".

**Institutional Review Board Statement:** Not applicable.

**Informed Consent Statement:** Not applicable.

**Conflicts of Interest:** The authors declare no conflict of interest.

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
