# Peer review of "Applications of Digital Twin across Industries: A Review"

_applsci, doi:10.3390/app12115727_

Round 1

Reviewer 1 Report

This review provides a comprehensive description of applications of Digital Twin in 13 different industries, including manufacturing, agriculture, education, construction, medicine, and retail, as well as industrial use cases in these industries.   The review is well written and can be easily understood by both professional and non-professional people.    It gives an accurate explanation and summary of relevant concepts in this paper.   Compared with other relevant reviews, this review has a comprehensive summary of DT column table description at the end, which is a very meaningful review for readers.

However, there is a problem with some segment formats, please check for modifications. For example, 13 education paragraph 2, 15 conclusion paragraph 1

After modifying the format of the manuscript, I recommend it be published.

Reviewer 2 Report

The presented paper reports a review on application for digital twin (DT). Specifically, it covers the applications of DT in 13 different industries.

In the opinion of this reviewer the paper is well written and treats a really  interesting topic. 

Major comments:
- it could be interesting to analyze also possible disadvantages related to the employment of DT such as security and privacy issues. Especially for healthcare application. 
- Are information regarding the computational cost and power consumption required to implement an accurate DT available in literature?

Minor comments:
- line 148: a space should be inserted after [55].
- please adapt the text style at lines 664-667 and 686-689 to the text format adopted for the whole paper.

Reviewer 3 Report

  1. Please correct the sequencing of references, they are not in ascending order.
  2. Line 85-87, needs paraphrasing as it is not establishing connection with the prior sentence.
  3. As this is a review-based study, figure 2 needs to be improved. Rather, then depicting information about the sectors DT is applied only, it should also give the number of studies being carried out in each sector. This will strengthen the review study and present a clearer picture for the researcher to visualize.
  4. There should be an overview in the introductory phase about the journals (its rankings, quartiles etc.) from where the studies have been picked.
  5. Please check the lines (781-789) and make necessary formatting corrections.
  6. In the conclusion section, the applications are discussed, however, future insights are missing.

Reviewer 4 Report

Dear author,

I have a few comments on the article:

  • Line 57 and 248 add space before reference.
  • Line 148 space before dot.
  • Line 326 to much space before.
  • Line 664-689 different font.
  • Line 672 dot is missing.
  • Line 781 aliment of the text isn’t according to the template.

Best regards, reviewer

Reviewer 5 Report

The authors have presented a survey on digital twin. In this survey, the applications of digital twin are described in detail. Along with this, the advantages are also discussed. The survey is presented nicely in my opinion. 

Reviewer 6 Report

The paper discusses the overview of the current research in the different fields of Digital Twins (DT) and their applications.

The subject is presented well. All the explanations are technically sound, attached figures are sufficient. References are sufficient. Acceptable in present form.

Round 2

Reviewer 3 Report

The modifications in the manuscript are carried out accordingly and the manuscript has improved.